

**Dust storms from the Taklamakan Desert significantly darken snow**
**surface on surrounding mountains**
Yuxuan Xing[1], Yang Chen[1], Shirui Yan[1], Tenglong Shi[1], Xiaoyi Cao[1], Xiaoying Niu[1],
Dongyou Wu[1], Jiecan Cui[1,2], Xin Wang[1,3], Wei Pu[1]
[1]Key Laboratory for Semi-Arid Climate Change of the Ministry of Education, College of Atmospheric
Sciences, Lanzhou University, Lanzhou 730000, China
[2]Zhejiang Development & Planning Institute, Hangzhou 310030, China
[3]Institute of Surface-Earth System Science, Tianjin University, Tianjin 300072, China
*Correspondence to*: Wei Pu (puwei@lzu.edu.cn)





**Abstract**

The Taklamakan Desert (TD) is a major source of mineral dust emissions into the atmosphere. These dust particles have the ability to darken the surface of snow on the surrounding high mountains after deposition, significantly impacting the regional radiation balance. However, previous field measurements have been unable to capture the effects of severe dust storms accurately, and their representation on regional scales has been inadequate. In this study, we propose a modified remote-sensing approach that combines data from the Moderate Resolution Imaging Spectroradiometer (MODIS) satellite and simulations from the Snow, Ice, and Aerosol Radiative (SNICAR) model. This approach allows us to detect and analyze the substantial snow darkening resulting from dust storm deposition. We focus on three typical dust events originating from the Taklamakan Desert and observe significant snow darkening over an area of >2100, >600, and >630 $km^2$ in the Tien Shan, Kunlun, and Qilian Mountains, respectively. Our findings reveal that the impact of dust storms extends beyond the local high mountains, reaching mountains located approximately 1000 km away from the source. Furthermore, we observe that dust storms not only darken the snowpack during the spring but also in the summer and autumn seasons, leading to increased absorption of solar radiation. Specifically, the snow albedo reduction (radiative forcing) triggered by severe dust depositions is up to 0.028–0.079 (11–31.5 W $m^{-2}$), 0.088–0.136 (31–49 W $m^{-2}$), and 0.092–0.153 (22–38 W $m^{-2}$) across the Tien Shan, Kunlun, and Qilian Mountains, respectively. This further contributes to the aging of the snow, as evidenced by the growth of snow grain size. Comparatively, the impact of persistent but relatively slow dust deposition over several months during non-event periods is significantly lower than that of individual dust event. This highlights the necessity of giving more attention to the influence of extreme events on the regional radiation balance. Through this study, we gain a deeper understanding of how a single dust event can affect the extensive snowpack and demonstrates the potential of employing satellite remote-sensing to monitor large-scale snow darkening.

**1 Introduction**



High Mountain Asia (HMA), which includes the Tibetan Plateau (TP) and surrounding mountain ranges, holds the largest amount of glaciers and snow outside of the poles. This region is informally known as the "The Third Pole" and the "Asian Water Tower" (Yao et al., 2012, 2019) because of its extreme importance as a freshwater source, with approximately one billion people relying on the water and hydropower that the glaciers and snow across HMA regularly provide (Immerzeel et al., 2012; Mishra et al., 2018). The snow-covered area of HMA is a highly reflective natural surface that has a significant impact on the regional radiation balance (Cohen and Rind, 1991; Painter et al., 2012). Previous satellite- and ground-based observations have demonstrated that the mass and extent of the snow cover across HMA are rapidly declining owing to recent global warming (Bormann et al., 2018; Notarnicola et al., 2020; Pulliainen et al., 2020). Furthermore, growing evidence has indicated that light-absorbing particles (LAPs), such as mineral dust and black carbon (BC), can induce snow darkening effect when they are deposited on the snow surface (Wang et al., 2013; Qian et al., 2015; Dang et al., 2017; Huang et al., 2022; Niu et al., 2022; Réveillet et al., 2022). This snow darkening effect increases solar absorption and decreases snow albedo, resulting in enhanced snowmelt and an imbalance in the Asian Water Tower (Hadley and Kirchstetter, 2012; Dumont et al., 2014; He et al., 2017, 2018; Shi et al., 2021, 2022a, 2022b; Cordero et al., 2022). Consequently, the snow-darkening effect plays a critical role in snow decline across HMA, thereby perturbing the climate system and impacting hydrological cycles (Kraaijenbrink et al., 2017, 2021; Sang et al., 2019; Shi et al., 2019; Zhang et al., 2020, 2021; Roychoudhury et al., 2022; Yang et al., 2022).

The Taklamakan Desert (TD) in southwestern Xinjiang, Northwest China, is the second-largest shifting sand desert on Earth and accounts for 42% of all dust emissions in East Asia (Chen et al., 2017a). Approximately 70.54 Tg of dust are emitted into the atmosphere annually, with the most intense dust events occurring in spring (Chen et al., 2017a). The dust in the Tarim Basin is predominantly redeposited onto nearby regions owing to the surrounding high mountains (Qiu et al., 2001; Sun et al., 2001; Shao and Dong, 2006). When the dust is uplifted above 4 km altitude, it may eventually settle on the snow surfaces across the surrounding high mountains, such as the Tien Shan and



Kunlun Mountains and subsequently induce a snow-darkening effect (Ge et al., 2014;
Jia et al., 2015; Yuan et al., 2018). Furthermore, this dust is also transported eastward
beyond the Tarim Basin and can be transported all the way to the Qilian Mountains via
the westerly winds during spring and summer, thereby inducing a snow darkening effect
in this distal region to the east of the TD (Dong et al., 2020; Han et al., 2022). Therefore,
TD dust may have a profound effect on the regional radiative balance by darkening the
snow across the high mountains surrounding the TD. This effect may subsequently
accelerate snow melting and affect water resources for the 30+ million people living in
the Xinjiang and Gansu provinces of China (Mishra et al., 2021).
Numerous field measurements have been undertaken in recent decades to investigate
the dust content of snow/glaciers across the high mountains surrounding the TD, with
measured dust contents generally varying from 1.4 to 110 µg g$^{-1}$ (Wake et al., 1994;
Dong et al., 2009, 2014; Wu et al., 2010; Ming et al., 2016; Xu et al., 2016; Schmale et
al., 2017; Zhang et al., 2018, 2021; Wang et al., 2019; Li et al., 2021, 2022). This
abundance of dust particles has been found to induce a significant snow darkening
effect across the high-mountain snowpack, thereby increasing its associated radiative
forcing to 25.8–65.7 W m$^{-2}$. Furthermore, the estimated natural dust-induced snow-
darkening effect can be equivalent to that induced by BC, particularly during intense
springtime dust events_(Sarangi et al., 2020; Zhang et al., 2021). These findings
effectively highlight the significance of the TD dust-induced snow darkening effect
across the surrounding high mountains. In spite of these invaluable in situ findings,
ground-based observations are poorly represented at the regional scale owing to limited
spatial coverage and temporal discontinuity. Furthermore, these previous field
measurements may not be able to capture severe dust emission and loading events,
which are more likely to induce snow darkening than common dry and wet deposition
processes (Dumont et al., 2020; Pu et al., 2021; Baladima et al., 2022).
Satellite remote sensing offers an effective way to overcome the limitations of ground-
based measurements by providing a more comprehensive understanding of the LAP-
induced impact on the regional radiative forcing of the snowpack (Skiles et al., 2018a).



For example, Painter et al. (2012) found that the instantaneous LAP-induced radiative
forcing can exceed 250 W m$^{-2}$ in the Hindu Kush-Himalaya region via an analysis of
Moderate Resolution Imaging Spectroradiometer (MODIS) satellite data. Sarangi et al.
(2020) further revealed that dust is the primary factor responsible for high-altitude snow
darkening in the Hindu Kush–Himalaya region. Similarly, severe dust events from the
Sahara can deposit dust on the snowpack across the European Alps and Caucasus
Mountains (Di Mauro et al., 2015; Dumont et al., 2020), with this deposition inducing
a radiative forcing of up to 153 W m$^{-2}$ based on satellite retrievals in Europe. Dust
deposition has also induced extensive snow darkening across the Upper Colorado River
Basin in North America, particularly during extreme dust events (Skiles et al., 2016,
2018b; Painter et al., 2017). These studies have demonstrated the effectiveness of
employing satellite remote sensing to estimate the dust content of the snowpack and its
associated radiative forcing. However, detecting natural dust deposition on the snow
surfaces across high mountains surrounding the TD is still limited.
Here we investigate the impact of dust storms on snow albedo reduction and radiative
forcing across the high mountains surrounding the TD. We first utilize MODIS satellite
data and the Snow, Ice, and Aerosol Radiative (SNICAR) model to retrieve the dust
content of the snowpack. We then capture three typical dust events that induced snow
darkening in the Tien Shan, Kunlun, and Qilian Mountains, respectively. Finally, we
analyze the spatial and altitudinal variations in dust-induced snow darkening and
compare our retrievals with field measurements. Through remote sensing observations,
we aim to provide a new view of the darkening effect of natural desert dust on the
snowpack of the high mountains surrounding the TD.
**2 Methodology**
**2.1 Remote-sensing data**
We accessed two MODIS datasets, the surface reflectance (MOD09GA:
https://earthdata.nasa.gov; 500 × 500 m resolution) and aerosol optical depth (AOD;
MCD19A2), to evaluate the impact of dust on snow albedo. MOD09GA is the daily
surface reflectance product from the Terra satellite, which provides the reflectance data





for seven bands (band 1, 620–670 nm; band 2, 841–876 nm; band 3, 459–479 nm; band
4, 545–565 nm; band 5, 1230–1250 nm; band 6, 1628–1652 nm; band 7, 2105–2155
nm). Previous studies have indicated that the MODIS sensor on Terra is not affected by
saturation on bright snow surfaces. As a result, it has the capability of detecting changes
in reflectance in the visible (VIS) bands caused by dust in snow (Painter et al., 2012;
Pu et al., 2019).
The daily averaged downward shortwave flux was obtained from the NASA Clouds
and the Earth's Radiant Energy System (CERES: https://ceres.larc.nasa.gov; 1° × 1°
resolution). The CERES data products take advantage of the synergy between
collocated CERES instruments and spectral imagers, such as MODIS (Terra and Aqua)
and the Visual Infrared Imaging Radiometer Suite (S-NPP and NOAA-20). We used the
downward shortwave flux to estimate the daily averaged radiative forcing that was due
to dust deposition on the snowpack. The Cloud-Aerosol Lidar with Orthogonal
Polarization (CALIOP/CALIPSO) provided by NASA is able to detect the type and
height of aerosols in the atmosphere (Huang et al., 2007; Han et al., 2022) and can
therefore be used to identify the movement of dust storms over the high mountains
surrounding the TD.
The Shuttle Radar Topography Mission (SRTM) digital elevation data, which possess
a 90-m spatial resolution, were provided by NASA and downloaded from Google Earth
Engine (https://earthengine.google.com). These data were used to correct the influence
of topography on surface reflectance.
**2.2 Snow depth and wind data**
The snow depth data were provided by NASA and accessed from the Modern-Era
Retrospective Analysis for Research and Applications, Version 2 (MERRA-2:
https://gmao.gsfc.nasa.gov). The MERRA-2 snow depth product was selected because
it has better accuracy than those from ERA-Interim, JJA-55, and ERA5 across HMA
(Orsolini et al, 2019). The wind field data were obtained from the European Centre for
Medium-Range Weather Forecasts (ECMWF) Reanalysis v5 (ERA5:
https://www.ecmwf.int) owing to its superior performance in terms of its high spatial
resolution and longer time span compared with other products (Copernicus Climate



Change Service, 2017). Here, we used ERA5 wind data at 700 hPa to describe the
atmospheric circulation during the analyzed dust storms.
**2.3 Radiative-transfer model**
The SNICAR model is a two-stream radiative transfer model (Flanner et al., 2007, 2009)
that has been widely used to simulate the spectral albedo of LAP-contaminated snow
(Sarangi et al., 2019; Chen et al., 2021). The model includes snow properties such as
snow depth and effective radius and accounts for the incident radiation at the surface
and its spectral distribution, solar zenith angle, and the type and concentration of LAPs
in the snowpack. In this study, dust optical parameters are taken from SNICAR defaults,
where the refractive index is 1.56 + 0.0038i at 0.63 μm (Patterson et al., 1981). And a
diameter bin of 0.1-1 μm was selected according to the previous observations from
Taklamakan Desert (Okada and Kai, 2004).
The Santa Barbara DISORT Atmospheric Radiative Transfer (SBDART) model is one
of the most widely used models for simulating the surface solar irradiance in clear and
cloudy sky conditions (Ricchiazzi et al., 1998). The SBDART model includes standard
atmospheric models, cloud models, extraterrestrial source spectra, gas absorption
models, standard aerosol models, and surface models. Here, we used the SBDART
model to calculate the spectral surface solar irradiance, following the approach of Cui
et al. (2021).
**2.4 Terrain correction**
The high mountains surrounding the TD have a complex terrain, such that the local
solar zenith angle ($\beta$) may differ from the MODIS-derived solar zenith angle ($\theta_0$).
Therefore, the topographic correction method should be used to derive $\beta$ (Teillet et al.,
1982; Negi and Kokhanovsky, 2011):
$$\cos \beta = \cos \theta_0 \cos \theta_T + \sin \theta_0 \sin \theta_T \cos (\phi_0 - \phi_T), \tag{1}$$
where $\phi_0$ is the solar azimuth angle from MODIS, and $\theta_T$ and $\phi_T$ are the surface
slope and aspect from SRTM, respectively. We then replace $\theta_0$ with $\beta$ in subsequent
satellite retrievals.
**2.5 Snow properties retrieval**



The dust-contaminated spectral snow albedo is determined based on the dust content,
snow grain size, snow depth, and solar zenith angle (Wiscombe and Warren, 1980). The
dust content and snow depth primarily impact the snow albedo in the ultraviolet (UV)
and VIS wavelengths, with a much smaller effect on snow albedo in the near infrared
(NIR) wavelengths (Figure 1 and Figure S1). Conversely, the snow grain size and solar
zenith angle primarily impact the snow albedo in the NIR wavelengths. The solar zenith
angle and snow depth data are from MODIS Terra and MERRA-2, respectively. We
used the SNICAR model to derive the quantitative snow grain size and dust content
from the MODIS data. We then used the SBDART model to estimate the dust-induced
snow albedo reduction and radiative forcing.

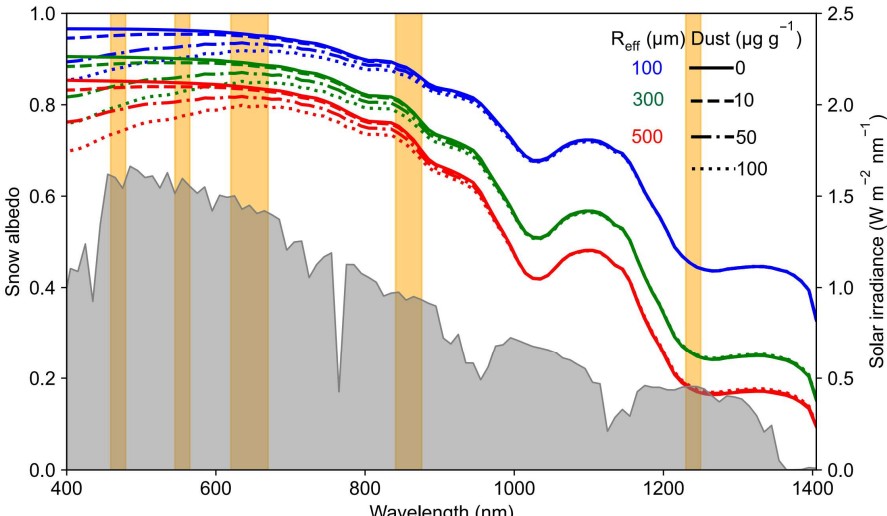


**Figure 1. Snow albedo spectra for different snow optical effective radius ($R_{eff}$) and**

**dust contents that were simulated using the SNICAR model. Orange bars denote**

**MODIS bands, and the gray region represents the typical solar irradiance in HMA.**

The Snow-Covered Area and Grain size (SCAG) model is a spectral unmixing method
that is widely used for identifying snow cover fraction (SCF) and snow optical effective
radius ($R_{eff}$), especially in complex mountain terrains (Painter et al., 2009, 2012; Rittger
et al., 2013). The SCAG model retrieves the SCF and $R_{eff}$ using all seven bands of the



MODIS reflectance data, which span the VIS to NIR range. It does not consider the
impact of LAPs. However, in our study, the dust content in snow is extreme high, which
will significantly reduce the VIS snow albedo in MODIS bands 1, 3, 4 (Figure 1). So,
the SCAG model will introduce a large bias in the resultant SCF and $R_{eff}$ retrievals.
Furthermore, the reflectance of fine-grained dirty snow has been compared with that of
pure coarse-grained snow at short-wave infrared wavelengths, which include bands 6
and 7 (Bair et al., 2020). The extremely high dust content in this study therefore means
that the reflectance in MODIS bands 6 and 7 is not appropriate for snow property
retrieval. Instead, we used the reflectance data in MODIS bands 2 and 5 to unmix the
surface reflectance to derive SCF and $R_{eff}$, similar to the approach in Painter et al.
(2009). The surface reflectance at band i ($R_{\text{band } i}^{\text{MODIS}}$) can be expressed as follows (Cui et
al., 2021):
$$R_{\text{band } i}^{\text{MODIS}} = \frac{E_{\text{band } i} \times \text{SCF} \times R_{\text{band } i}^{\text{MODIS, snow}} + E_{\text{band } i} \times (1-\text{SCF}) \times R_{\text{band } i}^{\text{soil}}}{E_{\text{band } i}}$$
$$= \text{SCF} \times R_{\text{band } i}^{\text{MODIS, snow}} + (1-\text{SCF}) \times R_{\text{band } i}^{\text{soil}}, \qquad (2)$$
where $R_{\text{band } i}^{\text{MODIS, snow}}$ and $R_{\text{band } i}^{\text{soil}}$ represent the snow and soil reflectances at band $i$,
respectively, with $R_{\text{band } i}^{\text{soil}}$ taken from Siegmund and Menz (2005), and $E_{\text{band } i}$ is the
solar irradiance at band $i$. The snow reflectance at band $i$ ($R_{\text{band } i}^{\text{MODIS, snow}}$) can be
expressed as
$$R_{\text{band } i}^{\text{MODIS, snow}} = \left( \frac{R_{\text{band } i}^{\text{MODIS}} - (1-\text{SCF}) \times R_{\text{band } i}^{\text{soil}}}{\text{SCF}} \right). \qquad (3)$$
We then fit the SNICAR-simulated snow reflectance to the MODIS-derived snow
reflectance, which is expressed as either
$$\text{RMSE} = \left( \frac{1}{2} (a \times (R_{\text{band } 2}^{\text{SNICAR, snow}} - R_{\text{band } 2}^{\text{MODIS, snow}})^2 + (R_{\text{band } 5}^{\text{SNICAR, snow}} - R_{\text{band } 5}^{\text{MODIS, snow}})^2 \right)^{\frac{1}{2}} \qquad (4)$$
or
$$\text{RMSE} = \left( \frac{1}{2} (a \times (R_{\text{band } 2}^{\text{SNICAR, snow}} - (\frac{R_{\text{band } 2}^{\text{MODIS}} - (1-\text{SCF}) \times R_{\text{band } 2}^{\text{soil}}}{\text{SCF}}))^2 \right.$$





$$+(R_{\text{band 5}}^{\text{SNICAR, snow}}-(\frac{R_{\text{band 5}}^{\text{MODIS}}-(1-\text{SCF})\times R_{\text{band 5}}^{\text{soil}}}{\text{SCF}}))^2)^{\frac{1}{2}},\qquad (5)$$
where RMSE is the root mean square error, $R_{\text{band }i}^{\text{SNICAR, snow}}$ is the SNICAR-simulated
snow reflectance at band $i$ (which is dependent on the $R_{\text{eff}}$ and solar zenith angle, where
the solar zenith angle is derived from the MODIS data), and $a$ is an empirical coefficient
(0.1–1 range). In this study, $a$ was set to 0.1 to reduce the interference of dust on the
snow properties retrieval because a high dust content can influence the snow albedo at
band 2 (Figure 1). We can then derive SCF and $R_{\text{eff}}$ by minimizing the RMSE (Painter
et al., 2009).

**2.6 Dust content and snow albedo reduction retrieval**

We fit the SNICAR-simulated snow reflectance to the MODIS-derived snow
reflectance in bands 3 and 4, which are the most sensitive to the dust content in snow,
following Pu et al. (2019) and Cui et al. (2021), which are expressed as either
$$\text{RMSE}=(\frac{1}{2}((R_{\text{band 3}}^{\text{SNICAR, snow}}-R_{\text{band 3}}^{\text{MODIS, snow}})^2+(R_{\text{band 4}}^{\text{SNICAR, snow}}-R_{\text{band 4}}^{\text{MODIS, snow}})^2))^{\frac{1}{2}}\qquad (6)$$
or
$$\text{RMSE}=(\frac{1}{2}((R_{\text{band 3}}^{\text{SNICAR, snow}}-(\frac{R_{\text{band 3}}^{\text{MODIS}}-(1-\text{SCF})\times R_{\text{band 3}}^{\text{soil}}}{\text{SCF}}))^2$$
$$+(R_{\text{band 4}}^{\text{SNICAR, snow}}-(\frac{R_{\text{band 4}}^{\text{MODIS}}-(1-\text{SCF})\times R_{\text{band 4}}^{\text{soil}}}{\text{SCF}}))^2)^{\frac{1}{2}},\qquad (7)$$
where $R_{\text{band 3}}^{\text{SNICAR, snow}}$ is a function of four factors: dust content, $R_{\text{eff}}$, snow depth, and
solar zenith angle. The latter three factors have been derived, leaving the dust content
as the only unknown. Therefore, the dust content can be retrieved by minimizing Eq.
(7). We assume that the derived dust content in this study accounts for the total light
absorption by all of the LAPs that are present in the snowpack. This is because our
study area is close to the Taklamakan Desert (TD), where large amounts of dust
accumulate on the snow surface annually. In contrast, anthropogenic activities and
biomass burning are rare, resulting in limited depositions of black carbon (BC) and
organic carbon (OC) (Fig. S8). Observations from snow and atmosphere have



confirmed this phenomenon (Wake et al., 1994; Huang et al., 2007). Therefore, our
assumption is plausible.
The dust-induced broadband albedo reduction ($\Delta\alpha$) can then be calculated as follows:
$$\Delta\alpha = \frac{\sum_{\lambda=300\text{nm}}^{\lambda=2500\text{nm}} E_\lambda \cdot \left( R_\lambda^{\text{SNICAR, pure-snow}} - R_\lambda^{\text{SNICAR, snow}} \right) \cdot \Delta\lambda}{\sum_{\lambda=300\text{nm}}^{\lambda=2500\text{nm}} E_\lambda \cdot \Delta\lambda}, \tag{8}$$

where $E_\lambda$ represents the total solar irradiance at wavelength $\lambda$ from the SBDART model,
$\Delta\lambda$ is 10 nm, and $R_\lambda^{\text{SNICAR, pure-snow}}$ and $R_\lambda^{\text{SNICAR, snow}}$ are the SNICAR-simulated pure
and polluted snow albedo, respectively. The dust-induced radiative forcing (RF) is
calculated as follows:
$\text{RF} = \Delta\alpha \cdot \text{SW}$,                                                        (9)
where SW is the downward shortwave flux, which is obtained from CERES.
The in situ dust content was not measured to verify the MODIS retrievals because of
the challenging geographical conditions surrounding the TD. Nevertheless, Cui et al.
(2021) verified a similar retrieval method and reported an uncertainty of less than ~40%
over highly polluted snow. As noted above, the snow albedo reduction is mainly
dependent on the dust content, $R_{\text{eff}}$, snow depth, and solar zenith angle. The $R_{\text{eff}}$ and
snow depth can be categorized as snow properties. We compared the dust content, snow
properties, and solar zenith angle to discuss their contributions to the spatial variations
in snow albedo reduction (Pu et al, 2019; Cui et al., 2021). The supplementary
information contains a thorough derivation of this method.
**3 Results**
**3.1 Remote sensing of the snow darkening effect across the high mountains**
**surrounding the TD**
The TD is located in the northern part of HMA and is surrounded by some of the highest
mountain ranges on Earth, including the Kunlun Mountains, Tien Shan, and Pamir
(Figures 2a and b). The TD region emits vast amounts of dust particles into the
atmosphere each year, particularly during the spring and summer (Wang et al., 2008;
Chen et al., 2013, 2017b; Kang et al., 2016; Wu et al., 2021; Tang et al., 2022); this
phenomenon is confirmed by the high AOD levels at 550 nm from March to August



(Figure 2c). A significant amount of this dust is ultimately redeposited across the Tarim
Basin and the surrounding mountains. The Tien Shan and Kunlun Mountains are two
regions that experience high levels of dust deposition owing to the local topography
and atmospheric circulation patterns (Figure 2d) (Huang et al., 2007, 2014; Ge et al.,
2014; Dong et al., 2022). Therefore, we selected two typical cases to demonstrate the
snow-darkening effect across the mountains surrounding the TD, a springtime dust
event across the Tien Shan and a summertime dust event across the Kunlun Mountains.

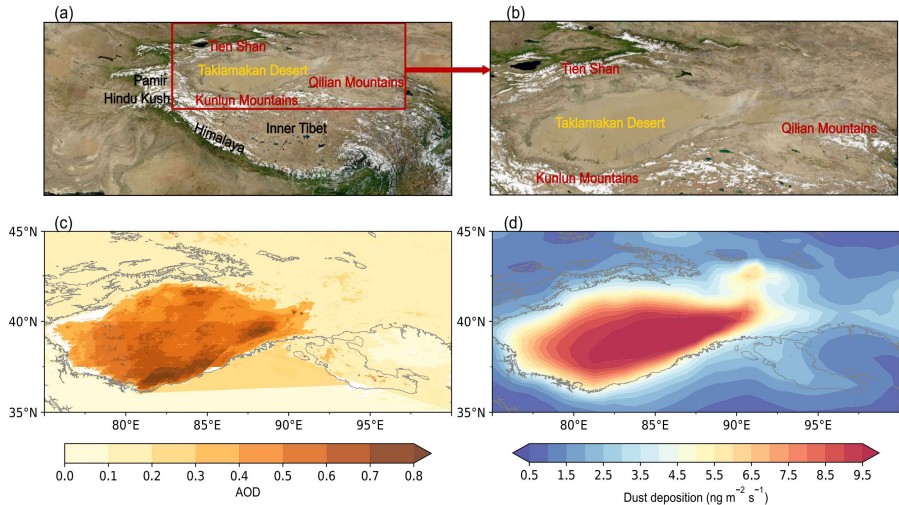

**Figure 2. Mountain ranges surrounding the Taklamakan Desert, and AOD and**
**dust deposition distributions across the Taklamakan Desert and surrounding**
**region. (a, b) Geographic location of the Taklamakan Desert and surrounding**
**mountains. The red box defines the area in (b). Spatial distributions of the**
**averaged (c) AOD and (d) dust deposition values, which were derived from**
**MCD19A2 and MERRA-2 during the March to August 2019 period.**
**3.1.1 Dust-induced snow darkening across the Tien Shan**
A significant dust storm occurred across the TD region on 18–22 May 2019. The 21
May 2019 Terra/MODIS satellite image (Figure 3b) showed that the dust plumes had
spread to the north and east owing to an upper anticyclone system in the Tarim Basin
(Figure 3h). Some dust particles were uplifted to >4 km altitude, as shown in the



CALIPSO aerosol vertical profiles (Figures 3j and k). These dust particles were then
transported to the snow-covered high-elevation areas of the Tien Shan, as illustrated in
the MODIS AOD images (Figures 3h and i). Dust plumes were also observed in a
satellite image that spanned the broadly snow-covered central Tien Shan (Figure 3e),
and the snow appeared to darken in the 22 May 2019 Terra/MODIS satellite image that
was acquired under the first clear-sky conditions after this severe dust event. However,
the snow was much whiter prior to the passage of this dust storm, as shown in Figures
3d and f. Figure 3g further illustrates changes in the surface reflectance of the snow-
covered areas, providing a more intuitive influence of dust deposition on the snow
physical properties. The reflectance was around 0.8 in the VIS spectrum on 15 May
2019, but quickly decreased to <0.7 on 22 May 2019, after the passage of the dust
plumes. The reduction in VIS wavelengths was up to >0.1 during this short time interval.
These observations show that the dust plumes from the TD can significantly darken the
snowpack across the Tien Shan through heavy dust deposition. Furthermore, the
progression of air-temperature-induced snow aging cannot effectively explain this
phenomenon. This result is consistent with previous satellite observations over the
Himalayas (Gautam et al., 2013).

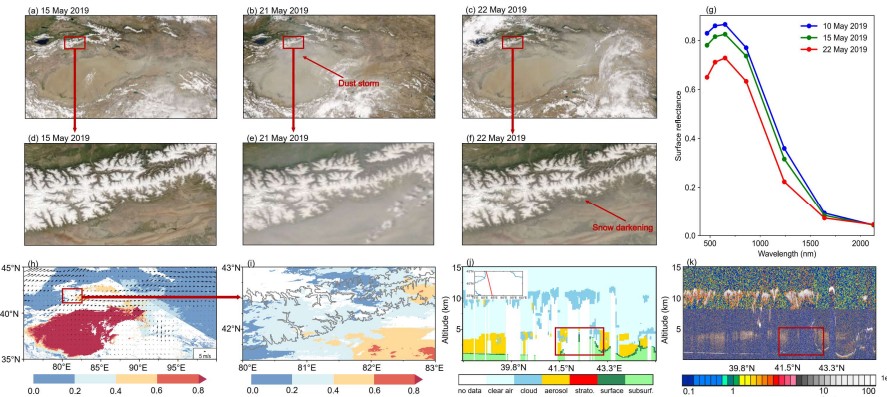


**Figure 3. Satellite observations during the 18–22 May 2019 severe dust event**
**across the Tien Shan. (a, d) Terra/MODIS satellite true-color images acquired on**
**15 May 2019, prior to the dust storm. (b, e) Terra/MODIS satellite images**
**acquired on 21 May 2019, with the dust storm transport from the TD to the Tien**



**Shan indicated by the red arrow in (b). (c, f) Terra/MODIS satellite images**
**acquired on 22 May 2019, with significant snow darkening observed across the**
**Tien Shan after the dust storm. (g) MOD09GA spectral surface reflectance across**
**snow-covered areas on 10 May 2019 (blue), 15 May 2019 (green), and 22 May 2019**
**(red). (h) MODIS AOD image on 21 May 2019, with the ERA5 daily mean wind**
**vector at 700 hPa overlain. (i) MODIS AOD image across the Tien Shan on 21**
**May 2019. Gray lines denote the 3000 m elevation contour. CALIPSO (j) vertical**
**feature mask and (k) backscatter coefficient on 21 May 2019.**
We also derived the spectral snow albedo and retrieved several parameters to
quantitatively assess the impact of this dust deposition on snow darkening. The
SNICAR-simulated spectral snow albedo (solid lines) and MODIS-derived 5-band
snow albedo (dots) in Figure 4a are averaged over the area in Figure 4c. These results
demonstrate an agreement of >95%, thereby indicating the reliability of our retrievals.
The spectral snow albedo reduction on 15 and 22 May 2019 are shown in Figure 4b.
There were significant increases in the albedo reductions as the wavelength decreased,
particularly on 22 May 2019, which is consistent with theoretical simulations of the
dust-induced snow darkening effect (Figure 1). However, the spectral curve differed
from the BC-induced results in the anthropogenically influenced areas of Northeast
China (Wang et al., 2017; Niu et al., 2022) and Northwest China (Shi et al., 2020).
Therefore, we indicate that the observed snow darkening in this study was mainly
caused by natural dust emissions, as opposed to BC and organic carbon (OC) emissions
from anthropogenic activities and/or biomass burning. There was a spectral snow
albedo reduction of 0.02–0.08 in the VIS on 15 May 2019, which represents persistent
but relatively low dust deposition during spring. However, the severe dust event caused
a rapid increase in spectral snow albedo reduction to 0.045–0.18 in a matter of days.
The approximate doubling of the albedo reduction indicates that the increase in the dust
concentration was much greater than 100% based on the nonlinear theory of the snow
albedo feedback to the dust concentration (Figure 1). This implies that it is important
to consider both the frequency and intensity of dust events when examining their impact





on snow albedo. Similar phenomena that were induced by catastrophic wildfire events have been observed in the snowpack across New Zealand (Pu et al., 2021). These results suggest that extreme events may reflect the more pronounced impact of climate warming on our planet (Liang et al., 2021; Gui et al., 2022). Therefore, it is important to pay more attention to extreme events, rather than just conducting either annual or monthly averaged analyses, to fully capture the influence of climate change on snow albedo.

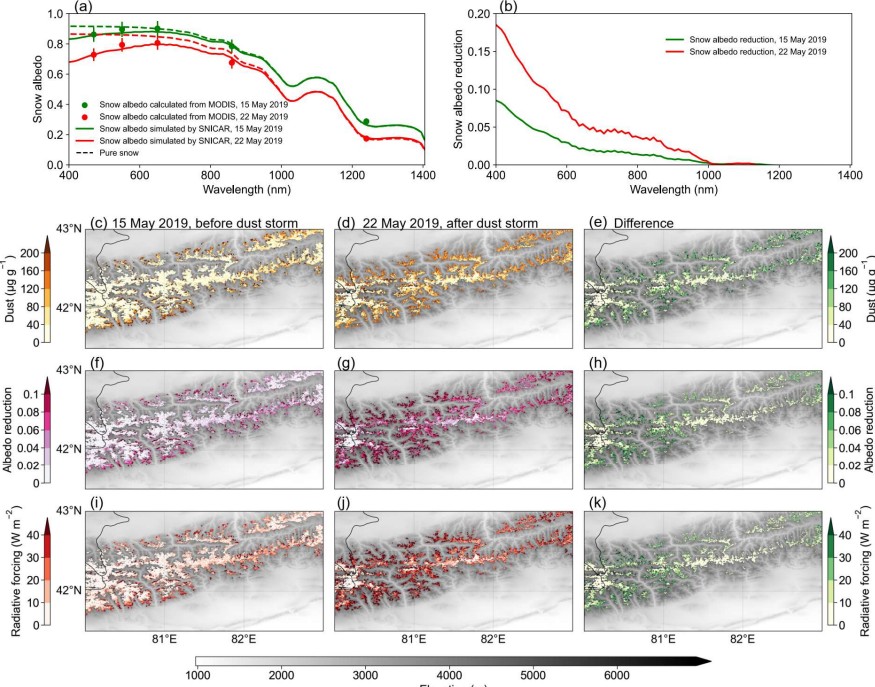

**Figure 4. (a) Averaged SNICAR-simulated spectral snow albedo (solid lines) and MODIS-derived 5-band snow albedo (dots) for the region across the Tien Shan impacted by the 18–22 May 2019 severe dust event. (b) Snow albedo reduction on 15 May 2019 (green) and 22 May 2019 (red). Spatial distributions of the average (c, d) dust, (f, g) albedo reduction, and (i, j) radiative forcing on 15 and 22 May 2019, respectively. Spatial distributions of the differences in (e) dust, (h) albedo reduction, and (k) radiative forcing between 15 and 22 May 2019. The background image in (c–k) is a grayscale topographic map of the Tien Shan.**



Figures 4c and d illustrate the spatial distributions of the dust concentration in the
snowpack on 15 and 22 May 2019, respectively. There was a sharp increase in the dust
content from 2–55 to 42–192 µg g$^{-1}$ (~2.67-fold increase) following the severe dust
event, with the lower elevations possessing higher dust concentrations and greater dust
content increases (Figures 4d and e). Snow darkening was observed across all of the
snow-covered areas (>2100 km$^2$), including the summits, thereby highlighting the
extensive influence of this severe dust event across the central Tien Shan. Furthermore,
these results demonstrate the capability and effectiveness of employing satellite remote
sensing to observe/monitor large-scale snow darkening. The dust-induced broadband
snow albedo reductions and radiative forcing are shown in Figures 4f–k, with observed
spatial patterns that are largely similar to the dust content distributions. The snow
albedo reduction increased by 0.008–0.052, with an observed increase from 0.002–
0.032 on 15 May to 0.028–0.079 on 22 May. The radiative forcing increased by 2.5–
20.5 W m$^{-2}$, with an observed increase from 0.5–12.5 W m$^{-2}$ on 15 May to 11–31.5 W
m$^{-2}$ on 22 May (Figure S7). Both the snow albedo reduction and radiative forcing
increased by a factor of ~2.39, which directly reflects its significant impact on the
regional radiation balance and climate (Dumont et al., 2020). Snow darkening can also
accelerate snow aging by absorbing more shortwave radiation in a warming spring, as
characterized by the $R_{eff}$ growth (Figures S3a–c).

**3.1.2 Dust-induced snow darkening across the Kunlun Mountains**


The Kunlun Mountains are located along the southern (northern) edge of the Tarim
Basin (Tibetan Plateau). The northern slope of the Central/West Kunlun Mountains
directly faces the TD (Figure 1a) and should have experienced the most severe dust-
induced snow darkening. Similar conditions also exist across the Himalayas, where the
south slope faces both the Thar Desert in India and the Middle East. We captured a
typical dust storm event with associated dust deposition and snow darkening that
occurred between 5 and 11 May 2020 along the northern slope of the Kunlun Mountains
using MODIS satellite images (Figure S2). The previously mentioned spring
phenomenon is well-known due to intense springtime dust emissions from the TD,



whereas the summer phenomenon is usually overlooked. However, it has been shown
that dust can more effectively cross the Kunlun Mountains during the summer months,
with the potential to induce changes in atmospheric dynamics and thermal effects (Yuan
et al., 2018). Therefore, we specifically chose a summer case to highlight snow
darkening across the Kunlun Mountains.

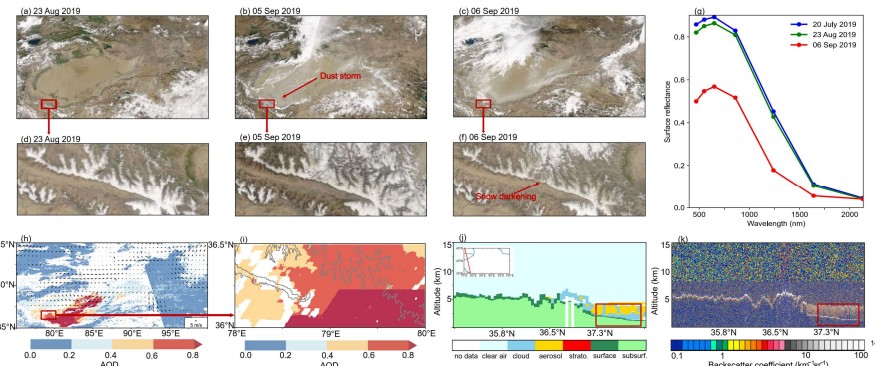


**Figure 5. Satellite observations during the 26 Aug to 08 Sep 2019 dust storm across**
**the Kunlun Mountains. (a, d) Terra/MODIS satellite true-color images acquired**
**on 23 Aug 2019, prior to the dust storm. (b, e) Terra/MODIS satellite images**
**acquired on 05 Sep 2019, with the dust storm transport from the TD to the Kunlun**
**Mountains indicated by the red arrow in (b). (c, f) Terra/MODIS satellite images**
**acquired on 06 Sep 2019, with significant snow darkening across the Kunlun**
**Mountains after the dust storm. (g) MOD09GA spectral surface reflectance over**
**the snow-covered areas on 20 July 2019 (blue), 23 Aug 2019 (green), and 06 Sep**
**2019 (red). (h) MODIS AOD image on 05 Sep 2019, with the ERA5 daily mean**
**wind vector at 700 hPa overlain. (i) MODIS AOD image across the Kunlun**
**Mountains on 05 Sep 2019. Gray lines denote the 3000-m elevation contour.**
**CALIPSO (j) vertical feature mask and (k) backscatter coefficient on 04 Sep 2019.**
A significant dust event that impacted the northern slope of the Kunlun Mountains
occurred from 26 Aug to 08 Sep 2019 (Figure 5b). The Terra/MODIS satellite images
on 5 Sep 2019 (Figures 5b and e) show the accumulation of dust plumes along the



southern edge of the Tarim Basin. In summer, the westerlies weaken and shift to the
north, leading to more accumulation of dust locally instead of transporting it eastward
(Chen et al., 2017a; Yuan et al., 2018). Furthermore, the enhanced sensible heat flux
favors the southward transport of uplifted dust, leading to cyclonic convergence at the
surface and anticyclonic divergence at the top of the troposphere above the TD (Figure
5h). The synergistic effects of atmospheric dynamic and thermal forcing can cause the
dust plumes to be uplifted to ~5 km altitude (Figures 5j–k). This uplift effectively
facilitated the dust plume ascent to the snow-covered areas across the northern slope of
the Kunlun Mountains (Figure 5e and i). A comparison of the MODIS images that were
acquired on 23 Aug and 6 Sep 2019 highlighted snow darkening after this severe dust
storm (Figures 5d and f). The surface reflectance decreased by ~0.22 in the VIS
spectrum, decreasing from 0.285 on 23 Aug to ~0.065 on 5 Sep. These observations
indicate that this summertime dust event caused significant snow darkening across the
Kunlun Mountains.

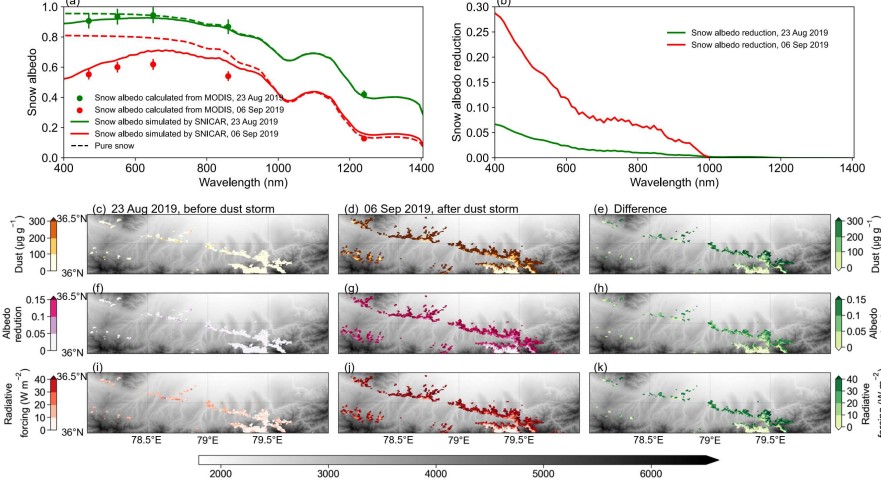


**Figure 6. (a) Averaged SNICAR-simulated spectral snow albedo (solid lines) and**
**MODIS-derived 5-band snow albedo (dots) for the region across the Kunlun**
**Mountains impacted by the 26 Aug to 08 Sep 2019 severe dust event. (b) Snow**
**albedo reductions on 23 Aug 2019 (green) and 06 Sep 2019 (red). Spatial**
**distributions of the average (c, d) dust, (f, g) albedo reduction, and (i, j) radiative**
**forcing on 23 Aug and 06 Sep 2019, respectively. Spatial distributions of the**





**differences in (e) dust, (h) albedo reduction, and (k) radiative forcing between 23 Aug and 06 Sep 2019. The background image in (c–k) is a grayscale topographic map of the Kunlun Mountains.**

Figure 6 provides a more quantitative investigation of the impact of this severe dust event on the snowpack across the Kunlun Mountains, whereby a significant increase in dust content from 12–50 µg g$^{-1}$ on 23 Aug to 170–360 µg g$^{-1}$ on 06 Sep (~6.45-fold increase) is observed after this severe dust event. The darkened snow-covered area spans >600 km$^2$, with a clear south–north gradient in the dust concentration distribution that is influenced by both the orientation and elevation of the mountains. This large dust deposition induced a 0.015–0.106 increase in snow albedo reduction, with an observed increase from 0.013–0.032 on 23 Aug to 0.088–0.136 on 06 Sep. There was also a substantial increase in radiative forcing of 4.1–37.5 W m$^{-2}$, with an observed increase from 3–11 W m$^{-2}$ on 23 Aug to 31–49 W m$^{-2}$ on 06 Sep (Figure S7). Note that these increases in both the snow albedo reduction and radiative forcing are approximately two times larger than those observed over the Tien Shan. These findings indicate accelerated snow aging, as evidenced by the faster growth rate of the R$_{eff}$ observed across the Kunlun Mountains (Figures S4 and S5).

**3.1.3 Snow darkening across the Qilian Mountains**

Unlike the Tien Shan and Kunlun Mountains, the Qilian Mountains are located approximately 1000 km east of the Tarim Basin. The Hexi Corridor, a narrow and relatively flat plain that lies between the high-elevation, inhospitable terrains of the Mongolian and Tibetan plateaus (see Figure 2), is situated to the north of the Qilian Mountains. The unique terrain of the region results in TD dust plumes following a preferred transport route across the Hexi Corridor to East Asia (Zhang et al., 2008; Meng et al., 2018). These dust plumes are generally uplifted to >4 km altitude and entrained in the westerlies (Huang et al., 2008; Dong et al., 2014; Chen et al., 2022), thereby providing a means for dust deposition onto the snowpack across the Qilian Mountains.





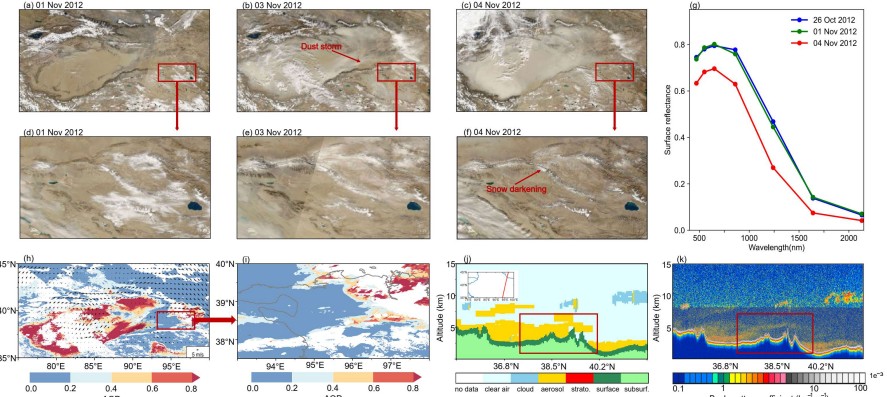

**Figure 7. Satellite observations during the 02–04 Nov 2012 dust storm across the Qilian Mountains. (a, d) Terra/MODIS satellite true-color images acquired on 01 Nov 2012, prior to the dust storm. (b, e) Terra/MODIS satellite images acquired on 03 Nov 2012, with the dust transport from the TD to the Qilian Mountains indicated by the red arrow in (b). (c, f) Terra/MODIS satellite images acquired on 04 Nov 2012, with significant snow darkening observed across the Qilian Mountains after the dust storm. (g) MOD09GA spectral surface reflectance over the snow-covered areas on 26 Oct 2012 (blue), 01 Nov 2012 (green), and 04 Nov 2012 (red). (h) MODIS AOD image on 03 Nov 2012, with the ERA5 daily mean wind vector at 700 hPa overlain. (i) MODIS AOD image across the Qilian Mountains on 03 Nov 2012. The gray line denotes the 3000-m elevation contour. CALIPSO (j) vertical feature mask and (k) backscatter coefficient on 03 Nov 2012.**

Figure 7 illustrates a severe dust event that occurred from 02 to 04 Nov 2012, when abundant dust plumes were being transported across the narrow Hexi Corridor (Figures 7b and h). The dust content was much more intense in this region, possessing AOD levels of up to >0.8. Furthermore, the CALIPSO observations indicated that the dust plumes were uplifted to ~10 km altitude (Figures 7j and k), thereby allowing some dust particles to cross over the northern slopes of the Qilian Mountains and spread across its western extent (Figures 7e and i). The average reflectance in the VIS spectrum was stable at around 0.7–0.8 across the snow-covered areas about a week before the severe

dust event but then significantly decreased to 0.6–0.7 owing to heavy dust deposition

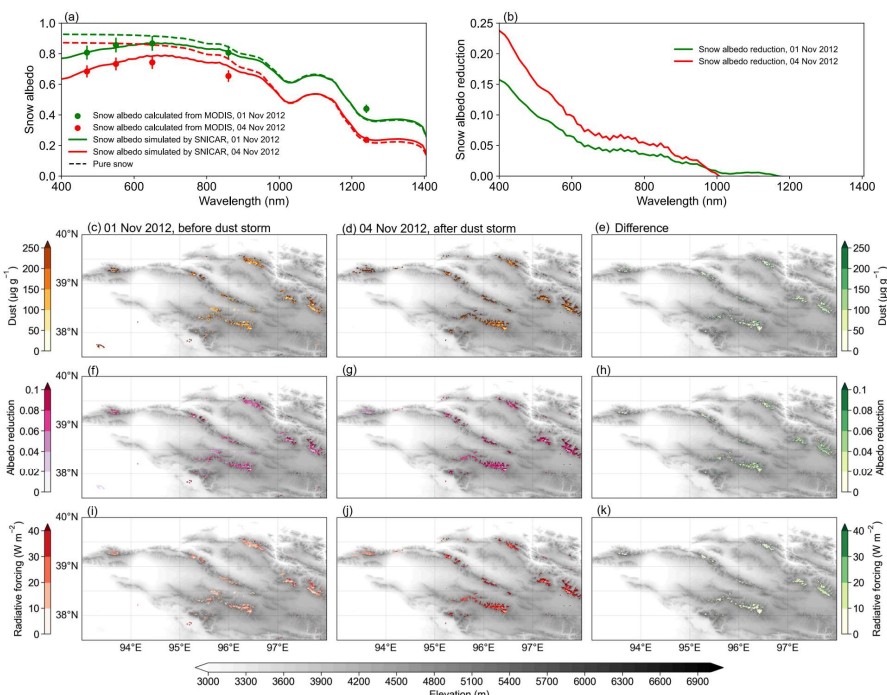

**Figure 8. (a) Averaged SNICAR-simulated spectral snow albedo (solid lines) and MODIS-derived 5-band snow albedo (dots) for the region across the Qilian Mountains impacted by the 02–04 Nov 2012 severe dust event. (b) Snow albedo reductions on 01 Nov 2012 (green) and 04 Nov 2012 (red). Spatial distributions of the average (c, d) dust, (f, g) albedo reduction, and (i, j) radiative forcing on 01 and 04 Nov 2012, respectively. Spatial distributions of the differences in (e) dust, (h) albedo reduction, and (k) radiative forcing between 01 and 04 Nov 2012. The background image in (c–k) is a grayscale image of the Qilian Mountains.**

Figure 8 presents the quantitative satellite-derived results, which highlight a rapid increase in dust content from 110–228 to 194–360 µg g$^{-1}$ (~1.53-fold increase) that spanned a snow-covered area of >630 km$^2$ (Figures 8f–h). This significant increase in dust content led to a considerable increase in snow albedo reduction (radiative forcing) of 0.018–0.067 (3–16 W m$^{-2}$), which increased from 0.042–0.076 (11–20 W m$^{-2}$) on 1



Nov 2012 to 0.092–0.153 (22–38 W m$^{-2}$) on 4 Nov 2012 (Figure S7). This >1.5-fold
increase in snow albedo reduction (radiative forcing) was not solely due to the
deposition of dust. Accelerated snow aging, which was observed from the enhanced
$R_{eff}$ growth (Figure S6), also contributed to the observed increase in snow albedo
reduction (radiative forcing); this trend was similar to that observed across the Kunlun
Mountains. Our approach uses satellite remote sensing to obtain a more complete
spatiotemporal evolution of the TD dust storm, including its emission, long-range
transport, and deposition, across the Qilian Mountains, which offers advantages over
previous field measurements (Wei et al., 2017).
**3.2 Contributions to the spatial and altitudinal variations in dust-induced snow**
**darkening**
We quantified the contributions of the three key factors (dust content, snow properties,
and solar zenith) to the spatial variations in snow albedo reduction (Figure 9) using the
method described in Section 2.6. The dust content was the dominant contributor to the
spatial variations in snow darkening. This is at least partially attributed to the greater
spatial differences in dust content compared with those of the other factors, as shown
in Figures 4, 6, and 8. Furthermore, theoretical modeling has indicated that the snow
albedo reduction is more sensitive to changes in dust content than to changes in the
snow properties and solar zenith angle (Flanner et al., 2021; Usha et al., 2022; Zhao et
al., 2022). Laboratory experiments also support these findings (Zhang et al., 2018; Li
et al., 2022). The contribution of the dust content also increased as the elevation in each
mountain range increased, whereas a decreasing trend was observed for the snow
parameters. This is because the dust content exhibits spatial differences across all of the
elevations owing to its widespread and heterogeneous depositions. However, the snow
depth has a more semi-infinite nature and $R_{eff}$ exhibits greater spatial homogeneity at
higher elevations owing to slower snow aging.









**Figure 9. Contributions of the spatial variations in dust content (blue), snow parameters (green), and solar zenith angle (red) to the snow albedo reduction at different elevations across the (a) Tien Shan, (b) Kunlun Mountains, and (c) Qilian Mountains.**

Scatter plots of the snow albedo reduction for the elevations across the Tien Shan, Kunlun Mountains, and Qilian Mountains are shown in Figure 10. The snow albedo reduction across the Tien Shan decreased with increasing elevation prior to the dust storm. However, the most severe dust deposition occurred within the 4000–4500 m elevation range, resulting in the most significant enhancement of snow albedo reduction in this elevation range. These findings are consistent with those reported for the Himalayas (Sarangi et al., 2020). The snow albedo reduction was generally low across the Kunlun Mountains for all of the elevation ranges. However, dust deposition caused the most significant albedo reduction within the 4500–5500 m elevation range, with a dramatic decrease of its influence above 6000 m. These findings correspond to the CALIPSO aerosol vertical profile observations (Figures 5j and k). The snow albedo reduction across the Qilian Mountains initially increased with elevation up to ~5000 m and then decreased at high elevations prior to the dust storm. However, the most severe dust deposition occurred across the lower elevations, leading to the most significant enhancement of snow albedo reduction across these lower-elevation regions. Our elevation analysis revealed a consistent outcome, whereby the dust storms significantly darkened the snowpack up to >5000 m elevation across the three analyzed mountain ranges.



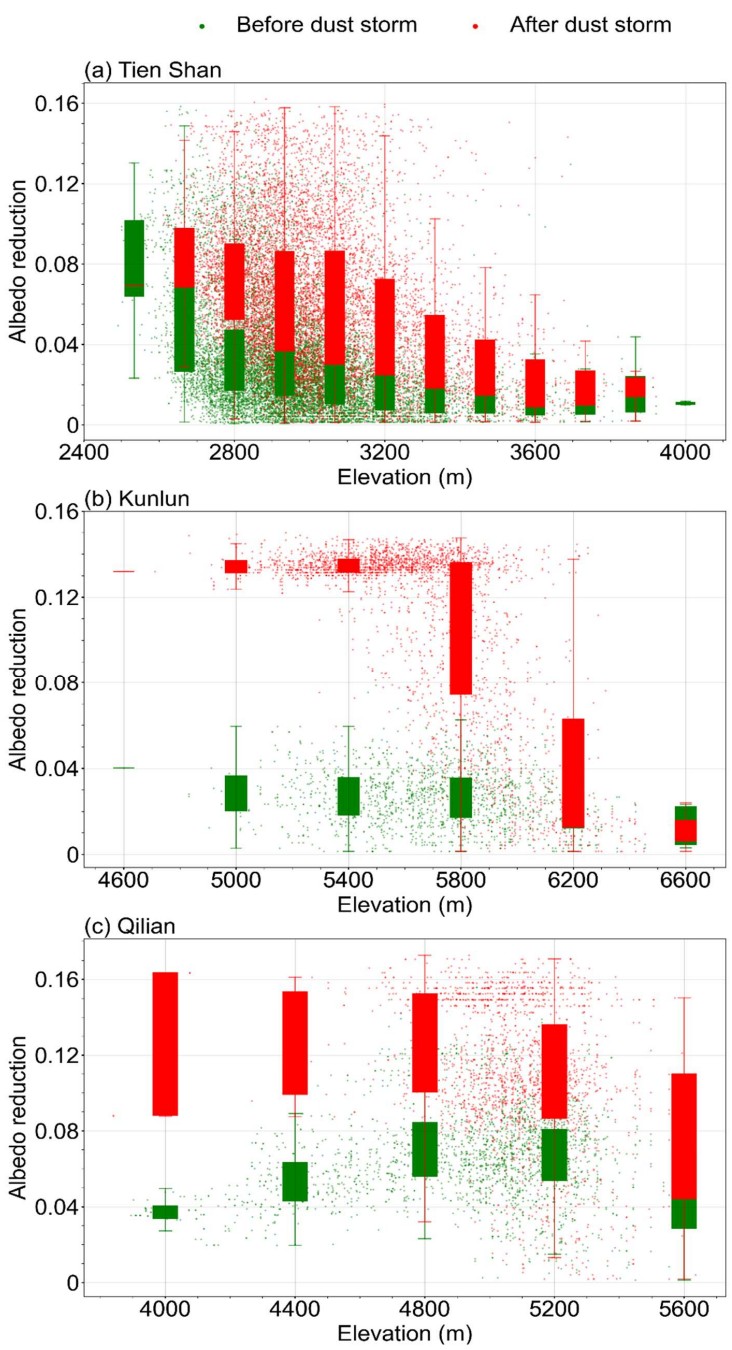

**Figure 10. Scatter plots of the snow albedo reductions for the analyzed elevation ranges across the (a) Tien Shan, (b) Kunlun Mountains, and (c) Qilian Mountains. Each box plot shows the statistical results for a 400-m elevation interval.**





**4 Discussion**

The snow darkening effect and its resultant radiative forcing have gained increasing attention in recent decades owing to their significant impacts on regional climate and hydrological systems. However, studies in the Tien Shan, Kunlun Mountains, and Qilian Mountains have been limited to local-scale observations, despite the significant impact of dust on snow darkening in these regions. Here we provide an overview of previous in situ dust-content measurements in the snowpack across the study region for comparison with our satellite remote-sensing results (see Figure 11). In the Tien Shan region, Ming et al. (2016), Xu et al. (2016), Li et al. (2021), and Zhang et al. (2021) reported a dust content of 19.3–110 µg g$^{-1}$ in the snowpack across Urumqi Glacier No.1. Dong et al. (2009) observed an average dust content of 0.97–3.69 µg g$^{-1}$ in the snowpack across Urumqi Glacier No. 1, Haxilegen Glacier No. 51, and Miaoergou Glacier. Schmale et al. (2017) found a variable dust content of 68.1–125.9 µg g$^{-1}$ in the snowpack across Suek Zapadniy, No. 354, and Golubin glaciers in the western Tien Shan. In the Kunlun Mountains, Wake et al. (1994) reported a dust content of up to ~8 µg g$^{-1}$ in the snow/ice across the western Kunlun Mountains. Wu et al. (2010) and Xu et al. (2016) measured dust contents of ~8.68 and 16.24 µg g$^{-1}$ in the ice core and snowpack across Muztagata Glacier in the northwestern Tibet Plateau (Wu et al., 2010; Xu et al., 2016), respectively. In the Qilian Mountains, Wu et al. (2010) analyzed ice cores from Dunde Glacier and measured a dust content of ~21 µg g$^{-1}$. The measured dust contents in the snowpack across Laohugou Glacier ranged from around 3 to 93.2 µg g$^{-1}$ (Dong et al., 2014; Xu et al., 2016; Zhang et al., 2018; Li et al., 2022). Wang et al. (2019) measured a variable dust content of 1.4–1.9 µg g$^{-1}$ in the fresh snow across Qiyi, Meikuang, and Yuzhufeng glaciers. Overall, previous field studies have reported dust contents of 0.97–125.9, 6.78–16.24, and 1.4–93.2 µg g$^{-1}$ for the Tien Shan, Kunlun Mountains, and Qilian Mountains, respectively.



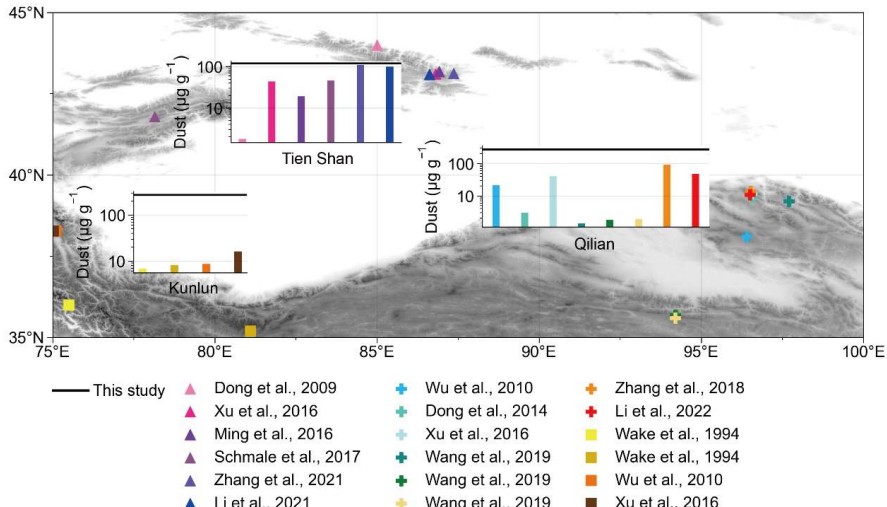

**Figure 11. Comparisons of the satellite-derived dust contents (black lines) in snow from this study and observed values from previous studies (colored symbols and bars).**

Our satellite-derived approach has yielded much higher dust contents than those obtained via in situ field measurements, with 42–196, 170–360, and 194–360 µg g$^{-1}$ determined for the Tien Shan, Kunlun Mountains, and Qilian Mountains, respectively. A key reason for this discrepancy could be that the field measurements usually record the background dust content signal, which includes a gradual natural deposition of dust, whereas our analysis specifically focused on significant snow darkening events due to severe dust storms, which further highlights the advantage of employing remote-sensing techniques to observe extreme snow darkening phenomena. We do note that satellite-derived approaches possess their own uncertainties, which arise from the data resolution and accuracy, algorithm assumptions, and atmospheric and underlying surface interferences (Cui et al., 2021). Nevertheless, this satellite-derived approach remains a valuable tool for effectively and rapidly studying extreme events, which cannot be captured by field measurements or climate model simulations, particularly as these extreme events will become increasingly important for climate and hydrological systems as the global climate continues to warm (Clow et al., 2016; Dumont et al., 2020).



Given the significant snow darkening effect highlighted in this study and recent
observations of decreasing snow cover across the Tien Shan, Kunlun Mountains, and
Qilian Mountains (She et al., 2015; Li et al., 2020; Zhu et al., 2022), it is crucial to
evaluate the impact of snow darkening on regional hydrologic cycles and local
freshwater supplies. However, snow aging and melting mechanisms are complex and
therefore require complementary observations because remote sensing alone cannot
distinguish the influences of augmented shortwave radiation owing to dust and
increased air temperatures on snow aging and melting (Gautam et al., 2013). Additional
research that integrates model simulations and satellite observations is necessary to
differentiate the roles of snow darkening and global warming in enhancing snow aging
and melting, and the resultant changes in glacier runoff in the future.

**5 Conclusions**

The Taklamakan Desert, the second-largest shifting sand desert on Earth, annually
emits vast amounts of dust into the atmosphere that eventually settles onto the
snowpack across the surrounding high mountains. We combined MODIS satellite data
analysis and SNICAR model simulations to reveal significant snow-darkening events
and quantify the snow albedo reduction and radiative forcing caused by severe dust
storms.
The satellite observations captured significant snow darkening over the 3000–6000 m
elevation range across the Tien Shan and Kunlun Mountains, which could be attributed
to the high uplift of dust owing to the local topography and atmospheric circulation.
The impacted area spanned the track of the dust storm and impacted almost all of the
snow-covered areas across the Tien Shan (>2100 km$^2$) and Kunlun Mountains (>600
km$^2$), including the summits. The dust content in the snowpack increased to 42–192
and 170–360 μg g$^{-1}$, with significant increases in snow albedo reduction (radiative
forcing) of 0.028–0.079 (11–31.5 W m$^{-2}$) and 0.088–0.136 (31–49 W m$^{-2}$) across the
Tien Shan and Kunlun Mountains, respectively. Furthermore, these dust events
accelerated snow aging, as indicated by the $R_{eff}$ growth. The dust plumes from the
Taklamakan Desert also traveled to the east, almost 1000 km from the Tarim Basin,



and deposited dust across much of the snow-covered area (>630 km$^2$) in the Qilian
Mountains. This dust deposition significantly increased the dust content to 194–360 μg
g$^{-1}$, causing a considerable increase in snow albedo reduction (radiative forcing) of
0.092–0.153 (22–38 W m$^{-2}$). The spatial distribution of the snow-darkening effect
varied across all three mountain ranges owing to the uneven deposition of dust.
Furthermore, the most significant snow darkening was observed in the high elevation
range (4000–5500 m). We also compared our satellite-derived results with previous
field measurements. Our results indicate that severe dust storms, which occur over short
periods, have a more profound effect on snow darkening compared with the relatively
slow deposition of dust when there are no dust storms. We therefore demonstrate that
satellite-derived analyses of dust deposition and its impact on snow albedo and radiative
forcing are crucial for rapidly and accurately capturing extreme dust deposition events
that may be difficult to detect through field measurements and climate model
simulations.













*Data availability*. All datasets and codes used to produce this study can be obtained by
contacting Wei Pu (puwei@lzu.edu.cn).
*Author contributions*. WP and XW designed the study and developed the overarching
research goals and aims. YX carried the study out and wrote the first draft with
contributions from all co-authors. YX processed the data with the assistance of YC, SY,
TS, XC, XN, DW and JC. WP and XW assumed oversight and leadership responsibility
for the research activity planning and execution. All authors contributed to the
improvement of results and revised the final paper.
*Competing interests*. The authors declare that they have no conflict of interest.
*Acknowledgements*. The Lanzhou University group acknowledges support from the
National Science Fund for Distinguished Young Scholars, the State Key Laboratory of
Cryosphere Science Open Fund and the National Natural Science Foundation of China.
We appreciate Dr. Boyuan Zhang's assistance with the code improvements. We thank
Lanzhou City's scientific research funding subsidy to Lanzhou University and the
Supercomputing Center of Lanzhou University for providing the computing services.
*Financial support*. This research was supported by the National Science Fund for
Distinguished Young Scholars (42025102), the State Key Laboratory of Cryosphere
Science Open Fund (SKLCS-OP-2021-05) and the National Natural Science
Foundation of China (42075061).



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
