# Peer review of "Dust storms from the Taklamakan Desert significantly darken snow"

_EGUsphere, 2023_

## Author Comment (AC1)

**Dear Editor Prof. Pedro Jimenez-Guerrero**,

Thank you very much for your time and effort in handling our manuscript entitled '***Dust storms from the Taklamakan Desert significantly darken snow surface on surrounding mountains***' (MS No.: egusphere-2023-1443). We appreciate your granting us additional time to improve the manuscript. We also appreciate your letters, which included questions and comments from the two reviewers as well as your valuable guidance.

The comments from the reviewers had great value and pointed out a number of flaws in our manuscript. We agree with the reviewers' comments and advice to improve our manuscript.

Now the major revision of the manuscript has been completed, we believe that the revised manuscript adequately addresses all reviewer concerns. Our point-to-point responses to the reviewer comments are attached in the files. We hope that the revised manuscript has been improved satisfactorily and that it can be accepted for publication in *Atmospheric Chemistry and Physics*.

If you feel that any deficiencies remain, please feel free to bring them to our attention. We would be grateful for any additional criticisms or suggestions.

Thank you again for your consideration and guidance. We look forward to hearing from you soon.

Yours faithfully,

Wei Pu

College of Atmospheric Sciences, Lanzhou University, Gansu, China

23 February, 2024

Response to Anonymous Referee #1

Your work is very interesting, but I have a serious concern. You haven't discussed the potential uncertainties and how they might influence your conclusions.

R: Thank you for your interest and constructive comments. We have supplemented the discussion on potential uncertainties in the revision. In response to the review comments, we have submitted a revised manuscript with track changes enabled to clearly indicate the modifications made. The followings are our point-by-point responses to the comments. Our responses start with "R:".

Convincing dust-induced snow-darkening within three days using MODIS/CALIOP daily snapshots is quite tricky.

R: Daily images have been added to illustrate the entire period of snow darkening events occurring from 15 May to 22 May 2019, 23 August to 6 September 2019, and 1 November to 4 November 2012 in the Tien Shan, Kunlun, and Qilian Mountains (Figures S2, S7, S10). However, due to the impact of cloud on MODIS image and LAPs retrieval, three representative MODIS images which captured before, during, and after dust events were selected to elucidate the effects of dust events on snow darkening, consistent with the study of Pu et al. (2021).

[Figure]

Figure S2. Satellite observations during the 15–22 May 2019 severe dust event across the Tien Shan (a-h).

[Figure]

Figure S7. Satellite observations during the 23 August to 06 September 2019 severe dust event across the Kunlun Mountains (a-o).

[Figure]

Figure S10. Satellite observations during the 01–04 November 2012 severe dust event across the Qilian Mountains (a-d).

Wouldn't the diurnal variabilities that MODIS misses cause significant biases?

R: The diurnal variations missing in MODIS images does not result in significant biases. LAPs and the associated albedo reductions, retrieved at 10:30 AM local time (coinciding with the MODIS Terra satellite overpass), were used as proxies for daily averages, in accordance with Painter et al. (2012). This approximation was reasonable, given that the content of LAPs exhibited little variation over a diurnal cycle (Painter et al., 2009; Zege et al., 2011). Daily snow albedo variation is primarily due to changes in the solar zenith angle (Figure S1). Given that the solar zenith angle mainly affects snow albedo in NIR, with little impact on the VIS, the diurnal variation in LAPs-induced snow albedo reduction was also considered limited. The revised content has been added to the manuscript. (Lines 310-318)

Could the dust in the atmosphere introduce biases in the MODIS view of the surface?

R: Thank you for the insightful comments. The dust in the atmosphere can lead to

biases in the retrieval results. The uncertainty analysis added to the manuscript includes the uncertainties caused by atmospheric dust. Cui et al. (2021) verified a similar retrieval method across the Northern Hemisphere, and we referenced their work to update the discussion on atmospheric dust uncertainty. The related analysis has been added in Lines 294-309 in revised manuscript as follow:

*"Cui et al. (2021) verified a similar retrieval method across the Northern Hemisphere. They considered that the accuracy of MODIS surface reflectance is typically ± (0.005 + 0.05 × reflectance) under conditions where aerosol optical depth (AOD) is less than 5.0, and solar zenith angle is less than 75°, as stated in the MODIS Surface Reflectance user's guide (Collection 6; https://modis.gsfc.nasa.gov/data/dataprod/mod09.php, last access: 19 January, 2024). In addition, the bias for snow grain size retrieval was assumed to be 30 % according to the studies of Pu et al. (2019) and Wang et al. (2017). These biases led to an overall uncertainty ranging from 10% to 110% in the retrieval of LAPs across the Northern Hemisphere. The study revealed that uncertainty decreased as LAPs concentration increased, with reported uncertainties dropping to below approximately 30% in regions of high pollution, such as Northeast China. In our study, the snowpack was also significantly polluted due to severe dust depositions, leading us to consider a retrieval uncertainty of 30% for LAPs, in alignment with the findings of Cui et al. (2021). Then, the overall lower bound and upper bound of the uncertainty value of snow albedo reduction retrieval was calculated and will be discussed in the following section."*

The overall lower bound and upper bound of the uncertainty value of snow albedo reduction retrieval has been added in Figure 5b, Figure 7b, Figure 9b, and Figure S5.

[Figure]

Figure 5. (a) Averaged SNICAR-simulated spectral snow albedo (solid lines) and MODIS-derived 5-band snow albedo (dots) for the region across the Tien Shan impacted by the 18–22 May 2019 severe dust event. (b) Snow albedo reduction on 15 May 2019 (green) and 22 May 2019 (red). Shadows indicate the retrieval uncertainty. Spatial distributions of the average (c, d) dust, (f, g) albedo reduction, and (i, j) radiative forcing on 15 and 22 May 2019, respectively. Spatial distributions of the differences in (e) dust, (h) albedo reduction, and (k) radiative forcing between 15 and 22 May 2019. The background image in (c–k) is a grayscale topographic map of the Tien Shan.

[Figure]

Figure 7. (a) Averaged SNICAR-simulated spectral snow albedo (solid lines) and MODIS-derived 5-band snow albedo (dots) for the region across the Kunlun Mountains impacted by the 26 Aug to 08 Sep 2019 severe dust event. (b) Snow albedo reductions on 23 Aug 2019 (green) and 06 Sep 2019 (red). Shadows indicate the retrieval uncertainty. Spatial distributions of the average (c, d) dust, (f, g) albedo reduction, and (i, j) radiative forcing on 23 Aug and 06 Sep 2019, respectively. Spatial distributions of the differences in (e) dust, (h) albedo reduction, and (k) radiative forcing between 23 Aug and 06 Sep 2019. The background image in (c–k) is a grayscale topographic map of the Kunlun Mountains.

[Figure]

Figure 9. (a) Averaged SNICAR-simulated spectral snow albedo (solid lines) and MODIS-derived 5-band snow albedo (dots) for the region across the Qilian Mountains impacted by the 02–04 Nov 2012 severe dust event. (b) Snow albedo reductions on 01 Nov 2012 (green) and 04 Nov 2012 (red). Shadows indicate the retrieval uncertainty. Spatial distributions of the average (c, d) dust, (f, g) albedo reduction, and (i, j) radiative forcing on 01 and 04 Nov 2012, respectively. Spatial distributions of the differences in (e) dust, (h) albedo reduction, and (k) radiative forcing between 01 and 04 Nov 2012. The background image in (c–k) is a grayscale image of the Qilian Mountains.

[Figure]

Figure S5. The overall lower bound and upper bound of the uncertainty value of snow albedo reduction retrieval due to atmospheric correction in Tien Shan (a-d), Kunlun Mountains (e-h) and Qilian Mountains (i-l).

Considering that CALIOP's track is merely a line over a MODIS granule, might assuming vertical profiles and aerosol types along the CALIOP track for the entire MODIS domain introduce biases in your analysis?

R: We used MODIS images to show the process of dust events. CALIOP solely tracked the dust height and was not involved in the retrieval process, thus it will not bring potential biases to the retrieval results. Additionally, Rohde et al. (2023) highlighted that aerosols in dust events predominantly consist of dust, as opposed to other aerosol types.

It would be helpful if you could add a section summarizing, and if possible, quantifying these uncertainties.

R: Thank you for your constructive comments. We have quantified the potential uncertainties in Section 2.6 and analyzed the overall lower bound and upper bound of the uncertainty value of snow albedo reduction retrieval in Section 3.1. Please refer to the revised manuscript for details.

1. Consider adding a map to indicate the regions you are referring to, especially in the introduction part. You could perhaps zoom out Figure 2(a).

R: Thank you for the good suggestion! We have added a map to indicate the region.

[Figure]

2. Good job listing a comprehensive set of references.

R: Thanks for your encouragement.

3. Since MODIS observes only once or twice per day, are you using simulations to minimize biases due to such sparse observations? (It seems like you're using models to "retrieve the dust content of the snowpack.")

R: Simulations were not employed; rather, LAPs and their associated albedo reductions, retrieved at 10:30 AM local time (MODIS Terra satellite overpass), served as proxy for daily averages in accordance with Painter et al. (2012). The reasonableness of this simplification has been discussed in the revised manuscript.

4. How significant is the diurnal variation of snow albedo for your estimations? It's possible that dust-induced darkening exhibits robust diurnal variability, which could introduce significant biases into your estimations.

R: The diurnal variations missing in MODIS images does not result in significant biases. LAPs and the associated albedo reductions, retrieved at 10:30 AM local time (coinciding with the MODIS Terra satellite overpass), were used as proxies for daily averages, in accordance with Painter et al. (2012). This approximation was reasonable, given that the content of LAPs exhibited little variation over a diurnal cycle (Painter

et al., 2009; Zege et al., 2011). Daily snow albedo variation is primarily due to changes in the solar zenith angle (Figure S1). Given that the solar zenith angle mainly affects snow albedo in NIR, with little impact on the VIS, the diurnal variation in LAPs-induced snow albedo reduction was also considered limited. The revised content has been added to the manuscript. (Lines 310-318)

5. Regarding CALIOP, it's important to clarify if you assumed the type of aerosol and its vertical profile to be the same across the entire MODIS image.

R: We used MODIS images to show the process of dust events. CALIOP solely tracked the dust height and was not involved in the retrieval process, thus it will not bring potential biases to the retrieval results. Additionally, Rohde et al. (2023) highlighted that aerosols in dust events predominantly consist of dust, as opposed to other aerosol types.

6. In the section discussing the radiative transfer models, you mention two models: one for obtaining contaminated snow and the other for simulating atmospheric radiative transfer. However, you don't explain how these two models are used or combined. This section needs more clarity.

R: To clarify the whole retrieval process, a flowchart delineating the step-by-step derivation of dust content, including snow albedo reduction and radiative forcing, has been incorporated as Figure 2. Eq. (8) integrates two models to compute dust-induced broadband albedo reduction, drawing on spectral snow albedo from SNICAR and spectral solar irradiance from SBDART simulations. It is consistent with Pu et al. (2019) and Cui et al. (2021). We have refined this content in Section 2.6 for improved clarity and augmented it with additional details on the spectral solar irradiance simulations using SBDART.

[Figure]

Figure 2. Flowchart illustrating the step-by-step retrieval of dust content and the associated snow albedo reduction and radiative forcing: the pink boxes denote the external input data, while the yellow boxes are used for calculations in this study.

7. Lines 161-163 mention that the SNICAR model provides spectral albedo, but lines 190-192 suggest you used the same model to derive snow grain size and dust content. This is confusing and should be clarified.

R: Sorry for the coarse description. To clarify the whole retrieval process, a flowchart illustrating the step-by-step derivation of dust content and the associated snow albedo reduction and radiative forcing has been added (Figure 2). We fit the SNICAR-

simulated snow reflectance to the MODIS-derived snow reflectance to retrieve snow grain size and dust content. According to Eq. (4)-(7), snow grain size is initially retrieved using MODIS reflectance in Bands 2 and 5. Subsequently, dust content is determined using MODIS reflectance in Bands 3 and 4. Finally, we use SNICAR to simulate spectral albedo across 350-2500 nm, utilizing the previously retrieved snow grain size and dust content.

8. In general, the radiative transfer part of your work lacks clarity and should be improved. In Section 3.1.1 (Figure 3) (and also in the other 2 examples), you aim to demonstrate dust-induced snow-darkening within three days using MODIS/CALIOP snapshots. It's important to address whether the surface reflectance product in MODIS could be affected by dust aerosols. Consider checking and showing CALIOP feature curtains (similar to your Figure 3j) for all three days to ensure that the darkening isn't due to atmospheric dust particles but rather snow-darkening.

R: Thank you for the insightful comment. We have improved the radiative transfer part of this work. We fully agree with your concern about the potential impact of atmospheric dust on the uncertainty of satellite retrieval results. We have updated the discussion on the uncertainty of snow albedo reduction, taking into account the effect of atmospheric dust on MODIS surface reflectance. Furthermore, in response to your comments, a comprehensive examination of all three-day CALIPSO images was conducted. However, it was found that, apart from the CALIPSO images already presented in our paper, the CALIPSO satellite did not pass over the study area during three dust events.

9. Line 23: Why >2100, >600,… km^2? Why can't put the approximated area?
R: We have revised as suggestion.

10. Line 35: 'Through' -> 'From'
R: We have revised as suggestion.

11. Line 48: satellite- -> satellite

R: We have revised as suggestion.

12. Line 56: I am not quite sure what 'imbalance' you are referring to here

R: Observational evidence and model projections that describe an imbalance in the Asian water tower caused by accelerated transformation of ice and snow into liquid water (Yao et al., 2022). In addition to climate change, light-absorbing particle deposition can also accelerate this process (Kang et al., 2020). We have revised the related content in the manuscript to clarify the expression.

**Reference**

Cui, J., Shi, T., Zhou, Y., Wu, D., Wang, X., and Pu, W.: Satellite-based radiative forcing by light-absorbing particles in snow across the Northern Hemisphere, Atmospheric Chemistry and Physics, 21, 269-288, 10.5194/acp-21-269-2021, 2021.

Kang, S., Zhang, Y., Qian, Y., and Wang, H.: A review of black carbon in snow and ice and its impact on the cryosphere, Earth-Science Reviews, 210, 10.1016/j.earscirev.2020.103346, 2020.

Pu, W., Cui, J., Shi, T., Zhang, X., He, C., and Wang, X.: The remote sensing of radiative forcing by light-absorbing particles (LAPs) in seasonal snow over northeastern China, Atmospheric Chemistry and Physics, 19, 9949-9968, 10.5194/acp-19-9949-2019, 2019.

Pu, W., Cui, J., Wu, D., Shi, T., Chen, Y., Xing, Y., Zhou, Y., and Wang, X.: Unprecedented snow darkening and melting in New Zealand due to 2019–2020 Australian wildfires, Fundamental Research, 1, 224-231, 10.1016/j.fmre.2021.04.001, 2021.

Painter, T. H., Rittger, K., McKenzie, C., Slaughter, P., Davis, R. E., and Dozier, J.: Retrieval of subpixel snow covered area, grain size, and albedo from MODIS, Remote Sensing of Environment, 113, 868–879, 10.1016/j.rse.2009.01.001, 2009.

Painter, T. H., Bryant, A. C., and Skiles, S. M.: Radiative forcing by light absorbing impurities in snow from MODIS surface reflectance data, Geophysical Research Letters, 39, n/a-n/a, 10.1029/2012gl052457, 2012.

Rohde, A., Vogel, H., Hoshyaripour, H. A., Kottmeier C., and Vogel, B.: Regional Impact of Snow-Darkening on Snow Pack and the Atmosphere During a Severe Saharan Dust Deposition Event in Eurasia, Journal of Geophysical Research: Earth Surface, 128, 10.1029/2022JF007016, 2023.

Wang, X., Pu, W., Ren, Y., Zhang, X., Zhang, X., Shi, J., Jin, H., Dai, M., and Chen, Q.: Observations and model simulations of snow albedo reduction in seasonal snow due to insoluble light-absorbing particles during 2014 Chinese survey, Atmospheric Chemistry and Physics, 17, 2279-2296, 10.5194/acp-17-2279-2017, 2017.

Yao, T., Bolch, T., Chen, D., Gao, J., Immerzeel, W., Piao, S., Su, F., Thompson., L., Wada, Y., Wang, L., Wang, T., Wu, G., Xu, B., Yang, W., Zhang, G., and Zhao, P.: The imbalance of the Asian water tower, Nature Reviews Earth & Environment, 3, 1-15, 10.1038/s43017-022-00299-4, 2022.

Zege, E. P., Katsev, I. L., Malinka, A. V., Prikhach, A. S., Heygster, G., and Wiebe, H.: Algorithm for retrieval of the effective snow grain size and pollution amount from satellite measurements, Remote Sensing of Enviroment, 115, 2674-2685, 10.1016/j.rse.2011.06.001, 2011.

Anonymous Referee #3

The manuscript titled "Dust storms from the Taklamakan Desert significantly darken snow surface on surrounding mountains" by Xing et al. focuses on the dust storm events from the Taklamakan desert and its influence on snow darkening processes over the distinct mountains in the High mountain Asia using the remote sensing techniques and modelling. This work is very interesting and novel. I have some major concerns and comments which can be incorporated during the revision process.

R: Thank you very much for the positive comments, which will encourage us to do more in-depth research in the future. Moreover, the referee's comments are quite significant that can help us to improve the paper quality substantially. We have addressed all of the comments carefully according to the suggestions. In response to the review comments, we have submitted a revised manuscript with track changes enabled to clearly indicate the modifications made. The followings are our point-by-point responses to the comments. Our responses start with "R:".

First of all, the models and the remote sensing data sets used in this study needs more clarification.

R: We have added more details to the description of the models and remote sensing datasets in Section 2.6 and Section 2.1.

The estimated radiative forcing needs more clarity. Several places authors mentioned that the dust content is derived using SNICAR model. It is confusing, and it needs further clarification and explanation.

R: A flowchart illustrating the step-by-step derivation of dust content and the associated snow albedo reduction and radiative forcing has been added to clarify the whole retrieval process (Figure 2). We fit the SNICAR-simulated snow reflectance to the MODIS-derived snow reflectance to retrieve dust content. According to Eq. (4)-(7), snow grain size is initially retrieved using MODIS reflectance in Bands 2 and 5. Subsequently, dust content is determined using MODIS reflectance in Bands 3 and 4.

[Figure]

Figure 2. Flowchart illustrating the step-by-step retrieval of dust content and the associated snow albedo reduction and radiative forcing: the pink boxes denote the external input data, while the yellow boxes are used for calculations in this study.

Also, the authors have used only very few cases in this study regarding the dust storm events, and some of the observations are from long ago. I would really recommend the authors to add some more dust storm event cases to this study.

R: Thanks for your suggestion. More dust event cases have been incorporated into our study to provide a more comprehensive analysis. Specific details have been added to the conclusion section, further indicating that severe dust events are not limited to the

three typical cases but are widely occurring.

Further, I would recommend the authors add in detail about the uncertainties of this study.

R: We have undertaken a comprehensive discussion on the potential uncertainties. Detailed discussions on uncertainty can be found in Section 2.6.

1.Line 41: Appropriate references for this statement. I would also recommend the authors to add the differences in the observations of light absorbing aerosols from the polar regions and HMA in the introduction section (for eg: Gogoi et al., 2021a, Gogoi et al., 2018, Chaubey et al., 2010 etc.).

M. Gogoi, S. Suresh Babu, Santosh K. Pandey, Vijayakumar S. Nair, et al., Scavenging ratio of black carbon in the Arctic and the Antarctic, Polar Science 16 (2018).

Gogoi, M.M., Pandey, S.K., Arun, B.S., Nair, V.S., Kumar, S., Vaishya, A., Prijith, S.S., Hegde, P., Babu, S.S., 2021b. Long-term changes in aerosol radiative properties over Ny-Ålesund : Results from Indian scientific expeditions to the Arctic. https://doi.org/10.1016/j.polar.2021.100700

Jai Prakash Chaubey, Krishna Moorthy, S. Suresh Babu, Vijayakumar S. Nair et al., Black carbon aerosols over coastal Antarctica and its scavenging by snow during the Southern Hemispheric summer, 2010. https://doi.org/10.1029/2009JD013381

R: We have added these references as suggestion.

2.Line 52: There are several studies from the Himalayan region, where higher concentrations of light-absorbing aerosols were reported from the atmosphere and snow. I would recommend using those references here. For eg: these are some of them.

Thakur, R.C., Arun, B.S., Gogoi, M.M., Thamban, M., Thayyen, R.J., Redkar, B.L., Babu, S.S., 2021. Multi-layer distribution of Black Carbon and Inorganic Ions in the Snow-packs of western Himalayas and Snow Albedo Forcing. Atmos Environ 261, 118564. https://doi.org/10.1016/j.atmosenv.2021.118564

Arun, B.S., Aswini, A.R., Gogoi, M.M., Hegde, P., Kumar Kompalli, S., Sharma, P., Suresh Babu, S., 2019. Physico-chemical and optical properties of aerosols at a background site (~4 km a.s.l.) in the western Himalayas. Atmos Environ 218, 117017. https://doi.org/10.1016/j.atmosenv.2019.117017

Arun, B.S., Gogoi, M.M., Borgohain, A., Hegde, P., Kundu, S.S., Babu, S.S., 2021. Role of sulphate and carbonaceous aerosols on the radiative effects of aerosols over a remote high-altitude site Lachung in the Eastern Himalayas. Atmos Res 263. https://doi.org/10.1016/j.atmosres.2021.105799

S. Arun, M. M. Gogoi, P. Hegde, A. Borgohain, Suresh K. R. Boreddy, S. S. Kundu, and S. S. Babu, 2021, Carbonaceous Aerosols over Lachung in the Eastern Himalayas: Primary Sources and Secondary Formation of Organic Aerosols in a Remote High-Altitude Environment, ACS Earth Space Chem. 2021. https://doi.org/10.1021/acsearthspacechem.1c00190

Gogoi, M.M., Babu, S.S., Arun, B.S., Moorthy, K.K., Ajay, A., Ajay, P., Suryavanshi, A., Borgohain, A., Guha, A., Shaikh, A., Pathak, B., Gharai, B., Ramasamy, B., Balakrishnaiah, G., Menon, H.B., Kuniyal, J.C., Srivastava, P., Singh, R.S., Kumar, R., Rastogi, S., 2021a. Response of ambient BC concentration across the Indian region to the nation-wide lockdown : results from the ARFINET measurements of 341–351.

R: Thank you for recommending the useful references. We have added these references as suggestion.

3.Line 81: I would recommend the authors to add more references for the atmospheric measurements of dust aerosols in the HMA region. For example; Arun et al., 2019 etc.

R: We have added reference as suggested.

4.Line 88: "_" correct it

R: The typographical error has been corrected.

5.Line 115: Using SNICAR, is it possible to retrieve the dust content? Needs clarification.

R: SNICAR can be used to retrieval the dust and other LAPs content (Pu et al., 2019; Bair et al., 2020; Cui et al., 2021). We have provided a detailed explanation on the method of SNICAR to retrieve dust content using a flowchart.

6.Line 123: Since the authors have used different remote sensing data sets with different spatial resolutions, I would recommend the authors to add the details regarding the uncertainties in this study related to it and add more details about the data products.

R: We have added as suggestion.

7.Line 163: I recommend the authors also go through the publication by Thakur et al., 2021 since it explains the multilayer distribution of LAP's in snow and snow albedo reduction.

R: We have added a discussion on the multilayer distribution of LAPs in snow. We have revised as suggestion. (Lines 181-185)

8.Line 166: This is not clear to me. And the cited reference seems too old in this case. SNICAR was released after 2000. Explain more about it."

R: We have added the related reference published by Flanner et al. (2007).

9.Line 171: How the authors have identified the clear and cloudy conditions during this study period? How did the authors combine these two data sets for the estimation?

R: Here, the images we use for satellite retrieval are cloud-free, so there are no situations with clouds. This condition is mainly used to calculate the solar irradiance provided by SBDART. For more details, you can refer to Eq. (2) in Cui et al., 2021. In this context, we can represent it with the following formula:

$$E_\lambda = E_{\mathrm{dif}}(\lambda; \varphi) + E_{\mathrm{dir}}(\lambda; \varphi) \times \cos\beta$$

where $E_\lambda$ represents the total solar irradiance at wavelength $\lambda$ from the SBDART model, $\varphi$ is latitude; $E_{\mathrm{dif}}(\lambda; \varphi)$ denotes the diffuse spectral irradiance on a horizontal

surface derived from the SBDART model under clear-sky conditions; and $E_{dir}(\lambda; \varphi)$ denotes the direct spectral irradiance on a horizontal surface derived from the SBDART model under clear-sky conditions. $\beta$ represents local solar zenith angle, which is obtained using the topographic correction method.

10.Line 194: How the authors have derived the snow grain size and dust content from the SNICAR model since this model is used to estimate albedo based on the user input of these parameters? Which needs clear clarification and explanation.

R: To clarify the whole retrieval process, a flowchart illustrating the step-by-step derivation of dust content and the associated snow albedo reduction and radiative forcing has been added (Figure 2). We fit the SNICAR-simulated snow reflectance to the MODIS-derived snow reflectance to retrieve snow grain size and dust content. According to Eq. (4)-(7), snow grain size is initially retrieved using MODIS reflectance in Bands 2 and 5. Subsequently, dust content is determined using MODIS reflectance in Bands 3 and 4. Finally, we use SNICAR to simulate spectral albedo across 350-2500 nm, utilizing the previously retrieved snow grain size and dust content.

11.Line 195: Explain in detail quantitatively about how the authors coupled the SNICAR and SBDART with all the atmospheric conditions in these two distinct models.

R: To clarify the whole retrieval process, a flowchart delineating the step-by-step derivation of dust content, including snow albedo reduction and radiative forcing, has been incorporated as Figure 2. Eq. (8) integrates two models to compute dust-induced broadband albedo reduction, drawing on spectral snow albedo from SNICAR and spectral solar irradiance from SBDART simulations. It is consistent with Pu et al. (2019) and Cui et al. (2021). We have refined this content for improved clarity and augmented it with additional details on the spectral solar irradiance simulations using SBDART.

12. Figure -1: Some of the labels are missing

R: Missing labels in Figure 1 have been added for clarity.

[Figure]

13. Line 209: Explain quantitatively about the bias

R: The bias you mentioned was the bias in the method of Painter et al. (2012), who did not give a quantitatively bias analysis. In our revised manuscript, we have added more details about the retrieval uncertainty in our method in Section 2.5 and Section 3.1.

14. Line 253: I would not completely agree with this assumption since the absorbing ability of BC is more compared to dust.

R: We agree with the reviewer's opinion that the absorption efficiency of BC is much higher than that of dust at the same concentration. However, studies have demonstrated that at higher altitudes, dust is the dominant factor contributing to snow darkening in the High Mountain Asia (Sarangi et al., 2020). Rohde et al. (2023) also indicated that dust is the primary aerosol type during dust events. Additionally, a rationale for this assumption is detailed in Lines 390-395. The appropriateness of this

assumption for investigating the influence of dust events on snow darkening is thus justified.

15.Line 287: Since the authors considered only two typical cases in this. I would recommend the authors to have some more multiple dust event cases in this study.

R: Thanks for your suggestion. More dust event cases have been incorporated into our study to provide a more comprehensive analysis (Figures S13-S21). Specific details have been added to the conclusion section, further indicating that severe dust events are not limited to the three typical cases but are widely occurring. We have revised the related content in the conclusion.

[Figure]

Figure S13. Satellite observations during the 29 September to 02 October 2015 severe dust event across the Tien Shan (a-f)

[Figure]

Figure S14. Satellite observations during the 01–06 March 2016 severe dust event across the Tien Shan (a-f).

[Figure]

Figure S15. Satellite observations during the 04–11 March 2023 severe dust event across the Tien Shan (a-f).

[Figure]

Figure S16. Satellite observations during the 28 January to 03 February 2019 severe dust event across the Kunlun Mountains (a-f).

[Figure]

Figure S17. Satellite observations during the 16–23 March 2019 severe dust event across the Kunlun Mountains (a-f).

[Figure]

Figure S18. Satellite observations during the 23–27 July 2019 severe dust event across the Kunlun Mountains (a-f).

[Figure]

Figure S19. Satellite observations during the 03–12 December 2014 severe dust event across the Qilian Mountains (a-f).

[Figure]

Figure S20. Satellite observations during the 25–30 December 2017 severe dust event across the Qilian Mountains (a-f).

[Figure]

Figure S21. Satellite observations during the 31 January to 03 February 2019 severe dust event across the Qilian Mountains (a-f).

16.Line 378: Explain the significance of diurnal variation of snow albedo for your estimation's biases in your estimations.

R: The diurnal variations missing in MODIS images does not result in significant biases. LAPs and the associated albedo reductions, retrieved at 10:30 AM local time (coinciding with the MODIS Terra satellite overpass), were used as proxies for daily averages, in accordance with Painter et al. (2012). This approximation was reasonable, given that the content of LAPs exhibited little variation over a diurnal cycle (Painter et al., 2009; Zege et al., 2011). Daily snow albedo variation is primarily due to changes in the solar zenith angle (Figure S1). Given that the solar zenith angle mainly affects snow albedo in NIR, with little impact on the VIS, the diurnal variation in LAPs-induced snow albedo reduction was also considered limited. The revised content has been added to the manuscript. (Lines 310-318)

17.Figure -3 , 5, 6 and 7: I would recommend to authors to make a clearer figure instead of keeping everything together.

R: Thank you for your suggestions. We have concentrated the information in the figures to make the manuscript's structure more concise. Furthermore, we have improved the clarity of these images.

18.Line 483: Why the authors have considered an old dust event in this study.

R: The consideration of cloud-free images is crucial in the quantitative assessment of the impact of Taklamakan Desert (TD) dust on snow darkening in the nearby mountains. It is noteworthy that dust events often occur in conjunction with cloudy conditions, which can compromise the reliability of the assessment. To address this, we chose to prioritize cloud-free images to ensure the accuracy of the analysis. Furthermore, recent dust event images have been included for reference (Figure S19, S20, S21).

19.Line 594: Give the references

R: The related reference has been added to support the statements made in this section.

20.Line 618: Conclusions in this study need to be concise and clear."

R: The conclusions section has been revised for conciseness and clarity, ensuring that the key findings and implications of our study are succinctly presented.

**Reference**

Arun, B. S., Aswini, A. R., Gogoi, M. M., Hegde, P., Kumar Kompalli, S., Sharma, P., and Suresh Babu, S.: Physico-chemical and optical properties of aerosols at a background site (~4 km a.s.l.) in the western Himalayas, Atmospheric Environment, 218, 10.1016/j.atmosenv.2019.117017, 2019.

Arun, B. S., Gogoi, M. M., Borgohain, A., Hegde, P., Kundu, S. S., and Babu, S. S.: Role of sulphate and carbonaceous aerosols on the radiative effects of aerosols over a remote high-altitude site Lachung in the Eastern Himalayas, Atmospheric Research, 263, 10.1016/j.atmosres.2021.105799, 2021a.

Arun, B. S., Gogoi, M. M., Hegde, P., Borgohain, A., Boreddy, S. K. R., Kundu, S. S., and Babu, S. S.: Carbonaceous Aerosols over Lachung in the Eastern Himalayas: Primary Sources and Secondary Formation of Organic Aerosols in a Remote High-Altitude Environment, ACS Earth and Space Chemistry, 5, 2493-2506, 10.1021/acsearthspacechem.1c00190, 2021b.

Bair, E. H., Stillinger, T., and Dozier, J.: Snow property inversion from remote sensing (SPIReS): A generalized multispectral unmixing approach with examples from MODIS and Landsat 8 OLI, IEEE Transactions on Geoscience and Remote Sensing, 59, 7270-7284, 10.1109/tgrs.2020.3040328, 2020.

Chaubey, J. P., Moorthy, K. K., Babu, S. S., Nair, V. S., and Tiwari, A.: Black carbon aerosols over coastal Antarctica and its scavenging by snow during the Southern Hemispheric summer, Journal of Geophysical Research: Atmospheres, 115, 10.1029/2009jd013381, 2010.

Cui, J., Shi, T., Zhou, Y., Wu, D., Wang, X., and Pu, W.: Satellite-based radiative forcing by light-absorbing particles in snow across the Northern Hemisphere, Atmospheric Chemistry and Physics, 21, 269-288, 10.5194/acp-21-269-2021, 2021.

Flanner, M. G., Zender, C. S., Randerson, J. T., and Rasch, P. J.: Present-day climate forcing and response from black carbon in snow, Journal of Geophysical Research, 112, 10.1029/2006jd008003, 2007.

Gogoi, M. M., Babu, S. S., Pandey, S. K., Nair, V. S., Vaishya, A., Girach, I. A., and Koushik, N.: Scavenging ratio of black carbon in the Arctic and the Antarctic, Polar Science, 16, 10-22, 10.1016/j.polar.2018.03.002, 2018.

Gogoi M. M., Babu, S. S., Arun, B. S., Moorthy, K. K., Ajay, A., Ajay, P., Suryavanshi, A., Borgohain, A., Guha, A., Shaikh, A., Pathak, B., Gharai, B., Ramasamy, B., Balakrishnaiah, G., Menon, H. B., Kuniyal, J. C., Krishnan, J., Gopal, K. R., Maheswari, M., Naja, M., Kaur, P., Bhuyan, P. K., Gupta, P., Singh, P., Srivastava, P., Singh, R. S., Kumar, R., Rastogi, S., Kundu, S. S., Kompalli, S. K., Panda, S.,  Rao, T. C., Das, T., and Kant, Y.: Response of ambient BC concentration across the Indian region to the nation-wide lockdown: results from the ARFINET measurements of ISRO-GBP, Current Science, 120, 10.18520/cs/v120/i2/341-351, 2021a.

Gogoi, M. M., Pandey, S. K., Arun, B. S., Nair, V. S., Thakur, R. C., Chaubey, J. P., Tiwari, A., Manoj, M. R., Kompalli, S. K., Vaishya, A., Prijith, S. S., Hegde, P., and Babu, S. S.: Long-term changes in aerosol radiative properties over Ny-

Ålesund: Results from Indian scientific expeditions to the Arctic, Polar Science, 30, 10.1016/j.polar.2021.100700, 2021b.

Li, Y., Chen, Y., and Li, Z.: Climate and topographic controls on snow phenology dynamics in the Tienshan Mountains, Central Asia, Atmospheric Research, 236, 10.1016/j.atmosres.2019.104813, 2020.

Pu, W., Cui, J., Shi, T., Zhang, X., He, C., and Wang, X.: The remote sensing of radiative forcing by light-absorbing particles (LAPs) in seasonal snow over northeastern China, Atmospheric Chemistry and Physics, 19, 9949-9968, 10.5194/acp-19-9949-2019, 2019.

Thakur, R. C., Arun, B. S., Gogoi, M. M., Thamban, M., Thayyen, R. J., Redkar, B. L., and Suresh Babu, S.: Multi-layer distribution of Black Carbon and inorganic ions in the snowpacks of western Himalayas and snow albedo forcing, Atmospheric Environment, 261, 10.1016/j.atmosenv.2021.118564, 2021.

Zege, E. P., Katsev, I. L., Malinka, A. V., Prikhach, A. S., Heygster, G., and Wiebe, H.: Algorithm for retrieval of the effective snow grain size and pollution amount from satellite measurements, Remote Sensing of Enviroment, 115, 2674-2685, 10.1016/j.rse.2011.06.001, 2011.